# Liquid-Plasma Hydrogenated Synthesis of Gray Titania with Engineered Surface Defects and Superior Photocatalytic Activity

**DOI:** 10.3390/nano10020342

**Published:** 2020-02-17

**Authors:** Feng Zhang, Guang Feng, Mengyun Hu, Yanwei Huang, Heping Zeng

**Affiliations:** 1State Key Laboratory of Precision Spectroscopy, East China Normal University, Shanghai 200062, China; 18843156039@163.com (F.Z.); myhu@phy.ecnu.edu.cn (M.H.); ywhuang@lps.ecnu.edu.cn (Y.H.); 2Shanghai Key Laboratory of Modern Optical System, Engineering Research Center of Optical Instrument and System, Ministry of Education, School of Optical-Electrical and Computer Engineering, University of Shanghai for Science and Technology, Shanghai 200093, China; sunnyfeng1992@163.com; 3Chongqing Institute of East China Normal University, Chongqing 401120, China; 4CAS Center for Excellence in Ultra-Intense Laser Science, Shanghai 201800, China

**Keywords:** surface defect, oxygen vacancy, gray titania, liquid plasma, visible light

## Abstract

Defect engineering in photocatalysts recently exhibits promising performances in solar-energy-driven reactions. However, defect engineering techniques developed so far rely on complicated synthesis processes and harsh experimental conditions, which seriously hinder its practical applications. In this work, we demonstrated a facile mass-production approach to synthesize gray titania with engineered surface defects. This technique just requires a simple liquid-plasma treatment under low temperature and atmospheric pressure. The in situ generation of hydrogen atoms caused by liquid plasma is responsible for hydrogenation of TiO_2_. Electron paramagnetic resonance (EPR) measurements confirm the existence of surface oxygen vacancies and Ti^3+^ species in gray TiO_2−x_. Both kinds of defects concentrations are well controllable and increase with the output plasma power. UV–Vis diffused reflectance spectra show that the bandgap of gray TiO_2−x_ is 2.9 eV. Due to its extended visible-light absorption and engineered surface defects, gray TiO_2−x_ exhibits superior visible-light photoactivity. Rhodamine B was used to evaluate the visible-light photodegradation performance, which shows that the removal rate constant of gray TiO_2−x_ reaches 0.126 min^−1^ and is 6.5 times of P25 TiO_2_. The surface defects produced by liquid-plasma hydrogenation are proved stable in air and water and could be a candidate hydrogenation strategy for other photocatalysts.

## 1. Introduction

Titanium dioxide (TiO_2_) due to its nontoxic, high stability, and high photocatalytic performance is considered as a promising photocatalyst, which has been widely used in environmental remedies and water splitting [1,2,3]. However, the photoactivity of TiO_2_ is largely limited by its wide bandgap (3.2 eV) and only responds to UV light, nearly 4% part of solar energy, which severely restricts its practical applications. In order to enhance solar light absorption, much effort has been devoted to bandgap engineering, by means of noble metal (Au, Pt, Ag, etc.) and nonmental (N, S, I, etc.) doping, artificial crystal structure modification, and carbonaceous nanomaterial compositing (carbon nanotubes, graphene, and fullerenes) [4,5,6,7,8,9,10]. Nevertheless, the visible-light energy conversion remains insufficient due to little solar light absorption and many carrier-recombination centers [11,12]. In recent years, black TiO_2_ has gained enormous attention owing to its extraordinary visible and infrared light absorption, prompting superior performances in photodegradation and water splitting [13,14,15]. The enhanced solar light absorption of black TiO_2_ is ascribed to additional intermediate electronic states (e.g., V_o_ or Ti^3+^ states) and disorder surface caused by hydrogenation [16,17,18]. More importantly, defect engineering, for instance defects concentration and distribution, also plays an important role in tuning photocatalytic activity of black TiO_2_. It was reported that the high defect ratio of surface to bulk can significantly enhance the photoactivity owing to preferential diffusion from bulk to surface of photoinduced charges [19,20]. However, surface defects are not stable enough as it can be spontaneously oxidized in air and water. So far, mainstream strategies to synthesize black/gray TiO_2−x_ are relied on the reduction of pristine stoichiometric TiO_2_ [20,21,22,23,24], such as annealing under high pressures of H_2_ or NH_3_, high-energy particle bombardment (hydrogen plasma, laser plasma, or high-energy electrons), and chemical reduction with reducing agent in vacuum. Apparently, it is significant and desirable to develop a simple and feasible strategy for massive production of black/gray TiO_2−x_ with engineered surface defects and related abundant solar absorption. 

Hereafter, we conduct a one-pot synthesis of gray TiO_2−x_ with large solar harvesting and engineered surface defects using a liquid-plasma technology at room temperature and atmospheric pressure [25]. There has recently been increasing interest in liquid-plasma discharges and their potential applications in various technologies, including environmental remediation, nanomaterial synthesis, and surface processing [26,27,28]. One of the most important advantages of liquid plasma is that various reactions instantaneously occur, including active species oxidation (e.g., OH, O, H_2_O_2_, etc.), ultraviolet radiation, electric fields, and shock waves. The synergy among those intense reactions is considered to possess a much higher efficiency than traditional chemical methods. In addition, liquid plasma provides an environmentally green and safe method, as fabrication itself consumes no extra chemicals and produces no chemical residuals.

We got mass production of black/gray TiO_2−x_ via liquid plasma driven by a bipolar pulse power. The synthesis setup, as shown in Figure 1A, is quite simple and it requires no complicated vacuum chambers, no reducing gas, or reagents. The bright liquid plasma was generated on the surface of metallic titanium electrode by applying high voltage pulses with high frequency as shown in Figure 1B. The typical generation procedure of liquid plasma can be detailed as follows. When appropriate pulse voltages were applied, some vapors were formed to cover the cathodes surface, i.e., “gas sheath”, by the Joule heating effect [29]. The “gas sheath” breakdown took place with the increase of electrolyte temperature and input voltage, and concurrently, glow discharge was generated near the cathodes surface. The temperature of cathode plasma was estimated to be 2000–3000 K according to previous studies [30,31]. Afterwards, cathodic plasma dissociated water molecules to produce the hydrogen environment at/near the plasma region [32]. After 1 h of liquid-plasma treatments, the color of 300 g anatase titania was varied from white to dark gray, and the digital pictures of dry powder samples are shown in Figure 1C. Figure 2 shows the optical emission spectrum of liquid plasma and clearly exhibits various emission peaks attributed to Ti I (neutral), Ti II (single-charged ions), hydroxyl radicals, hydrogen, and atomic oxygen, which were originated from the interaction between plasma and liquid. Thus, we hypothesize that white TiO_2_ nanoparticles dissolved in electrolyte entered the hot plasma area, and thus underwent plasma thermal treatment and hydrogenation at/near the plasma region. This synergy played a significant role in synthesizing gray TiO_2−x_, and the synthesis mechanism of gray TiO_2−x_ is depicted in Figure 2. From EPR measurements, a tunable surface-defect concentration of TiO_2−x_ was achieved by applying a different plasma discharge power. Interestingly, even though surface defects are easily oxidized under strong oxidation environments, these surface defects generated through the liquid-plasma strategy were verified as quite stable. 

## 2. Experimental Section 

### 2.1. Preparation of Gray TiO_2−x_

Commercial white anatase TiO_2_ nanopowders were bought from Shanghai Xieqing industrial Co., Ltd.(Shanghai, China), with 30 nm average size. P25 TiO_2_ of 20–30 nm was bought from Degussa Corp, which consisted of 78% anatase and 28% rutile. Briefly, we mixed 300 mL of 100 mg/mL well-dispersed anatase TiO_2_ solution and 7 mL of 0.1 mol/L nitric acid to get the electrolyte, which was afterwards transferred to electrolytic cell. One anodic platinum sheet (20 × 20 × 1 mm^3^, 99.99%) and two cathodic titanium rods (4-mm diameter, 99.9%) were sealed into a corundum tube in 300-mL nitric acid electrolyte as shown in Figure 1A. Two cathodes were used to generate glow discharges and avoid unbalanced flow and temperature gradients in the electrolyte. The HNO_3_ electrolyte here acted as a conductive solution. Pulsed voltages were applied between anode and cathodes to produce intense plasma nearby the cathode surfaces. Liquid plasma was produced on the cathode surface when we applied an appropriate pulse voltage power (600 V). The produced plasma whose intensity is dependent on glow discharge power, induces the high temperature at plasma–electrolyte interface and hydrogen generation. To avoid the electrolyte evaporation, a water chiller was used to maintain the electrolyte temperature at 80 °C. The color of TiO_2_ electrolyte gradually turned from white to dark gray within 1 h. The reaction mechanism for the formation of gray TiO_2−x_ can be ascribed to the hydrogenation effect of liquid plasma. Briefly, TiO_2_ nanopowders well dispersed in electrolyte took disordered movements due to intense liquid-plasma shockwaves. Some TiO_2_ nanoparticles moved to the plasma region and underwent hydrogenation under high temperature (2000–3000 K) owing to the hydrogen environment derived from water dissociation. Then, large numbers of nanoparticles repeated plasma hydrogenation treatments and hydrogenated gray TiO_2−x_ from white TiO_2_ was finally obtained. Hereafter, we dubbed the sample according to the output plasma power, for example GT-360 refers to gray TiO_2−x_ obtained by applying an output plasma power of 360 W.

### 2.2. Characterization

The phase and crystallinity for all samples were tested by the X-ray powder diffraction (XRD) using a Rigaku Smartlab (Rigaku, Tokyo, Japan) machine equipped with Cu Ka irradiation (λ = 1.54056 Å). The morphology was characterized by TEM using a JEM-2500SE (JEOL, Tokyo, Japan) instrument operating at an acceleration voltage of 200 kV. UV–Vis diffused reflectance spectra were measured by the Shimadzu UV-2700 spectrophotometer at a wavelength range of 200–800 nm at room temperature. The Raman spectra of samples were recorded by Thermo scientific DXR Raman microscope with 532-nm laser excitation at room temperature. The X-ray photoelectron spectra (XPS) were recorded with thermos Escalab 250Xi. The existence of defects doped in the TiO_2−x_ nanoparticles was confirmed by the X-band EPR spectra recorded at room temperature. The plasma discharges were controlled by applying square-wave voltages between titanium cathodes and Pt anode, with variable pulsed voltage of 0~800 V, pulsed current of 0~5 A, and repetition rate of 0~5 kHz. The voltage was increased at a ratio of 20 V/s.

### 2.3. Visible-Light Photocatalytic Degradation

The evaluation of visible-light photoactivity was individually tested with three typical waste-water pollutants, including rhodamine B (RhB), methyl orange (MO), and phenol. We used a 300-W Xenon lamp with a 420-nm cutoff filter as the visible-light source. Degussa P25 TiO_2_ was used as a standard photocatalyst for photoactivity comparison with as-prepared samples. The concentrations of RhB, MO, and phenol were 20, 20, and 10 mg/L, respectively. Firstly, 50 mg as-prepared TiO_2−x_ nanopowders and 50 mL of pollutant solution were put in a 500-mL beaker. Before illumination, the mixture was placed in a dark environment for 30 min with magnetic stirring for adsorption–desorption equilibrium. During photodegradation, we took out 1-mL of solution in the glass every 10 min and centrifuged it at 10,000 rpm for 11 min. Finally, the concentration of pollutant solution was analyzed at a specific wavelength by the UV-2700 spectrophotometer, where RhB, MO, and phenol are located at 554, 464, and 270 nm, respectively. Photodegradation experiments for whole tested samples were carried out under the same conditions.

## 3. Results and Discussion

The crystal structures of gray TiO_2−x_ obtained at different plasma discharge powers were detected by XRD analysis as shown in Figure 3. The XRD patterns of the as-grown gray TiO_2−x_ exhibit characteristic diffraction peaks matching the (101), (103), (004), (112), (200), (105), (211), (204), (116), (220), (215) facets of anatase crystal without any additional diffraction peaks, showing identical diffraction peaks with pristine TiO_2_. In addition, the full width at half maximum and height (FWMH), as well as the integrated peak area of the diffraction peak at 101 facets are nearly unchanged after plasma treatments as seen in Appendix A (see Support Information). This result indicates liquid plasma treatments neither destroy the long-range order lattice structure nor enlarge nanoparticle size of gray TiO_2−x_.

The morphology and lattice information of pristine TiO_2_ and gray TiO_2−x_ nanoparticles are presented in Figure 4A–D. Figure 4A shows that pristine TiO_2_ nanoparticles possessed serious agglomeration. Conversely, as shown in Figure 4B, little agglomeration was observed in gray TiO_2−x_, which could be ascribed to increasement of surface energy in nanoparticles owing to plasma treatments. Both pristine TiO_2_ and gray TiO_2−x_ nanocrystals are on average ~30 nm in diameter, which is well consistent with XRD measurements and suggests that hot temperature in liquid plasma does not prompt TiO_2_ crystal growth. Figure 4C,D present the high-resolution morphological structures of pristine TiO_2_ and TiO_2−x_ nanoparticles. From Figure 4C, marginal defects of pristine TiO_2_ are observed and could be produced during the crystal growth procedure. As indicated in Figure 4D, it seems that a thin disorder shell, nearly 1–2 nm in thickness, appears on the gray TiO_2−x_ surface, while the bulk maintains good crystallinity.

The UV–Vis diffuse reflectance spectra (DRS) are used to evaluate the light absorption properties of gray TiO_2−x_ samples. As shown in Figure 5A, pristine TiO_2_ shows merely ultraviolet light absorption, while all as-prepared gray TiO_2−x_ samples exhibit a broadband absorption from ultraviolet to visible even infrared regions. Moreover, from the inset of Figure 5A, the as-prepared TiO_2−x_ gradually exhibited a deepened color with the change of output power from 360 to 480 W. Figure 5B plots (αhν)^1/2^ versus hν by using the Kubelka–Munk function [33]. From which we calculate the bandgap of pristine TiO_2_, GT-360, GT-420, and GT-480 as 3.14, 3.07, 3.00, and 2.90 eV, respectively. Apparent red-shifts were observed for the absorption edge of gray TiO_2−x_ compared with pristine TiO_2_. The higher discharge power is, the narrower bandgap of gray TiO_2−x_ is obtained. The decreased bandgap in our case implies some self-doped crystal defects, for instance oxygen vacancy and/or Ti^3+^ centers, has been produced after plasma hydrogenation. Since self-doped defects inevitably introduce doped states between the bandgap such as oxygen-vacancy and Ti^3+^ induced intermedium states, which accounts for the visible-light absorption of gray TiO_2−x_. The color depth of defective TiO_2−x_ is reported and can be related to the amount of surface defects, which plays a role of “color centers”, and the more surface defects TiO_2−x_ possesses, the more visible light TiO_2−x_ captures [34,35]. To verify the existence of surface defects, gray TiO_2−x_ was heated at 300 °C under atmosphere for 3 h. Unexpectedly, the color was changed from gray to white, which can be attributed to the fact that surface defects diffuse from surface to bulk or can be oxidized by air. Thus, we conclude that some surface defects most likely exist in gray TiO_2−x_ and its quantity could be controlled directly by varying plasma discharge power.

The Raman spectra of gray TiO_2−x_ and pristine TiO_2_ are shown in Figure 6. Both samples exhibit similar Raman peaks at 142, 194, 395, 514, and 637 cm^−1^, respectively, which is ascribed to anatase TiO_2_, well consistent with the XRD results. It is observed that the E_g_ mode of anatase associated with the symmetric stretching vibration of O-Ti-O is shifted to higher wavenumbers compared with pristine TiO_2_, which reveals that the Ti-O bands environment is changed. In principle, this phenomenon can result from lattice defects and/or finite size effect (<10 nm). Our TEM measurements exclude the size effect as pristine TiO_2_ and gray TiO_2−x_ have almost the same nanoparticle size. Consequently, we infer that the difference between gray TiO_2−x_ and pristine TiO_2_ is mainly due to the lattice defects effect.

XPS analyses are further performed to investigate the surface chemical states of Ti and O in gray TiO_2−x_. Figure 7A shows the full-scale XPS spectra of pristine TiO_2_ and gray TiO_2−x_. No obvious differences are observed between pristine TiO_2_ and gray TiO_2−x_, indicating no other dopant elements in gray TiO_2−x_. Figure 7B shows a high-resolution Ti 2p XPS spectra for both samples. There are two main peaks for pristine TiO_2_ centered at a binding energy of 458.6 and 464.4 eV, which are ascribed to Ti^4+^ 2p_3/2_ and Ti^4+^ 2p_1/2_, respectively [36]. In comparison with pristine TiO_2_, two similar peaks of gray TiO_2−x_ are detected but with a slight shift toward lower binding energy at 458.3 and 464 eV, assigned to Ti^3+^ 2p_3/2_ and Ti^3+^ 2p_1/2_, respectively [37]. Figure 7C shows the XPS O 1s spectra for pristine TiO_2_ and gray TiO_2−x_. From the O 1s spectrum of pristine TiO_2_, two deconvoluted peaks at 529.6 and 531.4 eV are related to lattice oxygen (Ti-O) and surface absorption hydroxyl (Ti-OH), respectively [38]. Comparably, the fitting peaks of O 1s spectrum of gray TiO_2−x_ show an obvious shift toward higher binding energy at 529.7 and 531.5 eV, which is mainly due to the existence of oxygen vacancies. Moreover, the area of fitting peak of Ti-OH in gray TiO_2−x_ is little larger than that of pristine TiO_2_, indicating that many Ti dangling bonds induced by oxygen vacancies are produced. 

The existence of surface defects was confirmed by EPR spectra at room temperature as shown in Figure 8. Unexpectedly, the pristine TiO_2_ shows an apparent paramagnetic signal centered at g = 2.001 ascribed to surface oxygen defects, which is consistent with HRTEM measurement. In comparison with TiO_2_, gray TiO_2−x_ possesses two extinguished peaks at g = 2.001 and g = 1.974, corresponding to surface oxygen vacancies and surface Ti^3+^ species, respectively, which is in good accordance with XPS results [39,40]. As is well known, the formation of Ti^3+^ centers is always accompanied by the presence of oxygen vacancies due to the charge unbalance. It should be noticed that both of the EPR signals gradually increase with plasma discharge power. It is therefore proved that the high concentration of surface defects can be obtained when applying sufficient output energy. 

Based on the obtained results, the generation of surface defects in gray TiO_2−x_ could be understood that hydrogen atoms generated by liquid-plasma electrolysis reduced TiO_2_ to produce Ti^3+^ and oxygen vacancies. Furthermore, hydrogen atoms entered in the TiO_2−x_ lattice and some atoms bonded with Ti atoms while others bonded with O atoms, and thus created the surface lattice disorders [41]. Massive hydrogen atoms produced by enhanced output power create an intensified plasma hydrogenation effect, which induces the increased contents of surface Ti^3+^ and oxygen vacancies [42]. In addition, there is no color fade for gray TiO_2−x_ even if exposed in air and water for six months, indicating the existences of surface defects are quite stable. The robust surface defects could be attributed to the wrapped outer disorder surface with 1–2 nm thickness, hindering further strong oxidation by liquid plasma. On the other hand, considering the presence of surface Ti^3+^ and oxygen vacancies, the rather stable point defect structure of Ti^3+^-Vo-Ti^3+^ could be formed in order to maintain the electrostatic balance [43]. 

The photocatalytic performance for whole samples was tested with degradation of RhB under visible-light illumination (λ > 420 nm), as shown in Figure 9A. It is clearly seen that gray TiO_2−x_ shows a much higher photoactivity than that of pristine TiO_2_ and P25. Figure 9B shows the calculated ln(C_0_/C_t_) with the irradiation time (t), i.e., ln(C_0_/C_t_) = kt, where k is the apparent rate constant, which follows first-order kinetic behavior and can directly give the efficiency of photodegradation. The best performance of gray TiO_2−x_ is 0.126 min^−1^, which is far higher than pristine TiO_2_ and is 6.5 times of P25. In order to evaluate the stability of gray TiO_2−x_, the photocatalytic reaction was repeated four times under the same experimental conditions. As shown in Figure 9C, after four repeated cycles, the photocatalytic performance has no obvious differences, which indicates that the photoactivity of gray TiO_2−x_ is rather stable. Additionally, the visible-light degradation of methyl orange and phenol are shown in Figure 9D. After 2 h of photodegradation, the degradation rate of MO using gray TiO_2−x_ is 96.25%, while pristine TiO_2_ shows only 7.52%. The degradation rate of phenol for pristine TiO_2_ is 4.26%, while gray TiO_2−x_ shows a much higher rate than pristine TiO_2_ and reaches 89.2%. Furthermore, some hydrogenated titania and doped TiO_2_ with high performance in RhB photodegradation are provided for comparison with gray TiO_2−x_ as reflected in Table 1.

On basis of the above results, we can conclude that the superior photocatalytic activity of gray-TiO_2−x_ is mainly due to the existence of surface defects. It is generally accepted that photocatalytic activity is largely dependent on the quantity of absorbed photons. The more photons TiO_2_ can absorb, the more photoinduced electrons and holes it can produce. Owing to the existence of oxygen vacancies and Ti^3+^ centers, dopants states such as Ti^3+^ states and oxygen states appear between the bandgap, which can narrow the bandgap and allow more long-wavelength light absorption. Even though, lots of photoinduced e-h pairs yet are easily recombined in the bulk of TiO_2_ crystal, leading to a serious waste of photoinduced charges. Surface defects serve as holes absorbers and thus cause the directional diffusion of photogenerated e-h pairs to the TiO_2_ surface. Therefore, benefits of surface defects in gray TiO_2−x_ not only narrow its bandgap, but also reduce the recombination rate of e-h pairs. On the other hand, with the increase of output energy, the concentrations of surface defects are correspondingly improved, and lead to a narrower bandgap, as well as faster e-h pairs diffusion. It thus explains discharge power-dependent photoactivity of gray TiO_2−x_.

Considering the high temperature involved in liquid plasma treatments (2000–3000 K), the possible modification of black TiO_2_ deriving from the titanium cathodes should be discussed. Originally, we carried out the liquid plasma experiment to produce black TiO_2_, which was reported by others [16,17,18]. Typically, Ti cathodes (99.9%) and Pt anode (99.99%) were used and all other experimental conditions and setup are same with this work, except no TiO_2_ powder was mixed in the electrolyte. The black TiO_2_ nanoparticles were derived from Ti cathodes after plasma anodization and heat treatment, which shows abundant surface oxygen vacancies. However, the quantity of black TiO_2_ is quite few (about 10 mg), and it cannot meet our demands in the experiment and some practical applications. Furthermore, visible-light photodegradation of black TiO_2_ shows a heavy decrease in photocatalytic performance, which was ascribed to unstable existences of surface oxygen vacancies due to photocorrosion. To improve the quantity and photocatalytic performance, lots of anatase TiO_2_ powders were added and mixed in the electrolyte, which subsequently produced gray TiO_2−x_ with mass production and stable surface defects. Overall, the contamination of black TiO_2_ generated from Ti cathodes existed. Nevertheless, the stabilities of surface oxygen defects, as well as photodegradation performance exhibited in gray TiO_2−x_, which confirms that the mechanism of plasma power-related hydrogenation with TiO_2_ plays the major role in whole synthesis procedure. Therefore, contamination from Ti cathodes does not affect the performance of gray TiO_2−x_, and the modification can be ignored. On the other hand, platinum sheet as inert metals is resisted by heavy corrosion due to intense plasma anodization, and there is no mass decay of Pt anode after plasma treatments. Transmission electron microscopy-energy dispersive X-ray spectroscopy (TEM-EDS) was carried out to exclude the influence of Pt element as seen in Appendix A.

## 4. Conclusions

In summary, a simple green method assisted by liquid plasma has been demonstrated for the synthesis of gray TiO_2−x_ at a low temperature and atmospheric pressure. The synthesis mechanism can be explained that the in situ production of plasma-induced hydrogen atoms plays significant roles in the hydrogenation reactions of gray TiO_2−x_. The liquid-plasma technique can effectively manipulate the optical properties and band structures of gray TiO_2−x_, and herein enhances accompanying visible-light absorption. One interesting finding is that the engineered surface defects can be realized through directly controlling the plasma discharge power. EPR measurements confirm the existences of surface oxygen vacancies and Ti^3+^ species in gray TiO_2−x_. Both kinds of defects concentrations are well controllable and increase with the output plasma power. The synergistic effect with narrowed bandgap and directional e-h pairs diffusion results in superior visible-light photoactivity. Rhodamine B was used to evaluate the visible-light photodegradation performance. The results show that the removal rate constant of gray TiO_2−x_ reaches 0.126 min^−1^, and is 6.5 times of P25. Moreover, photodegradations of methyl orange and phenol confirm that gray TiO_2−x_ exhibits much higher photoactivity than pristine TiO_2_. We believe that this liquid plasma hydrogenation strategy can be extended to other photocatalysts with mass stable surface defects. Overall, our gray TiO_2−x_ with improved solar energy absorption possesses potential for more efficient polluted water cleaning.

## Figures and Tables

**Figure 1 nanomaterials-10-00342-f001:**
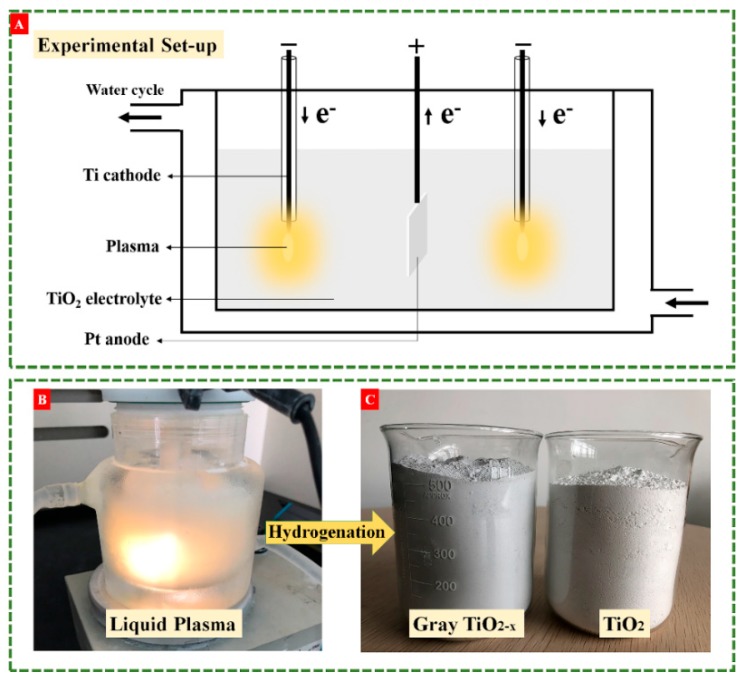
(**A**) The setup for liquid plasma generation, (**B**) digital picture of as-produced liquid plasma, and (**C**) digital pictures for gray titanium dioxide (TiO_2−x_) and pristine white TiO_2_.

**Figure 2 nanomaterials-10-00342-f002:**
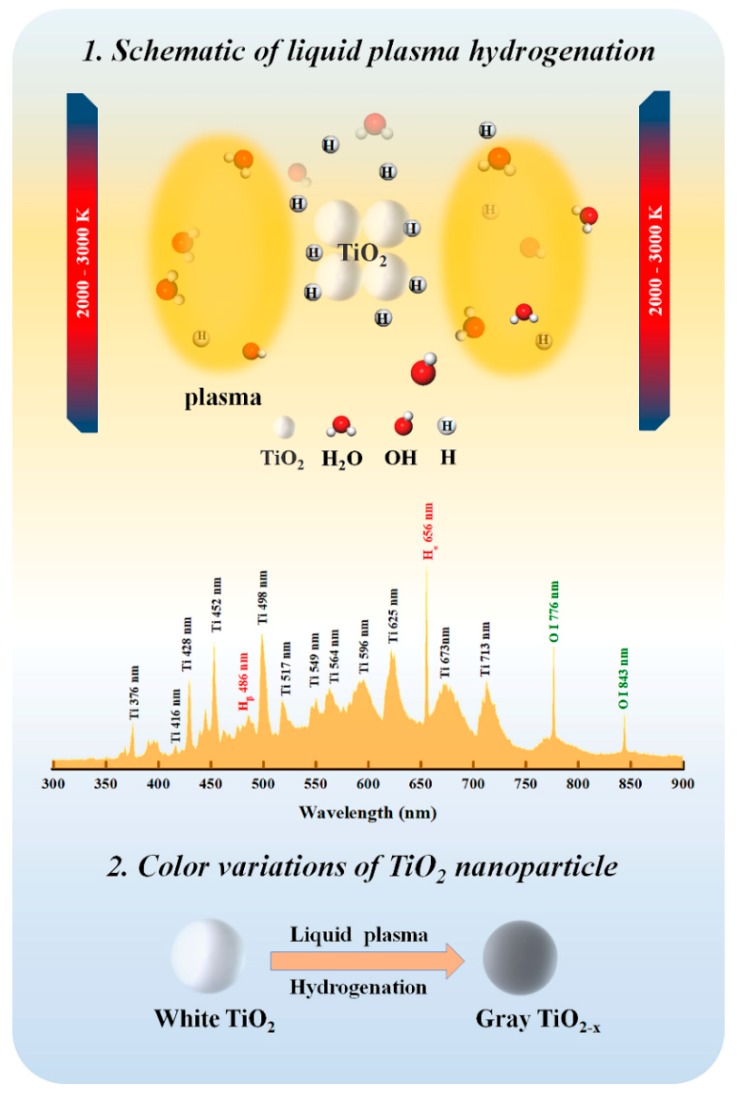
The synthesis mechanism of gray TiO_2−x_ nanosphere assisted by liquid plasma and the corresponding optical emission spectrum originated from cathodic Ti electrodes.

**Figure 3 nanomaterials-10-00342-f003:**
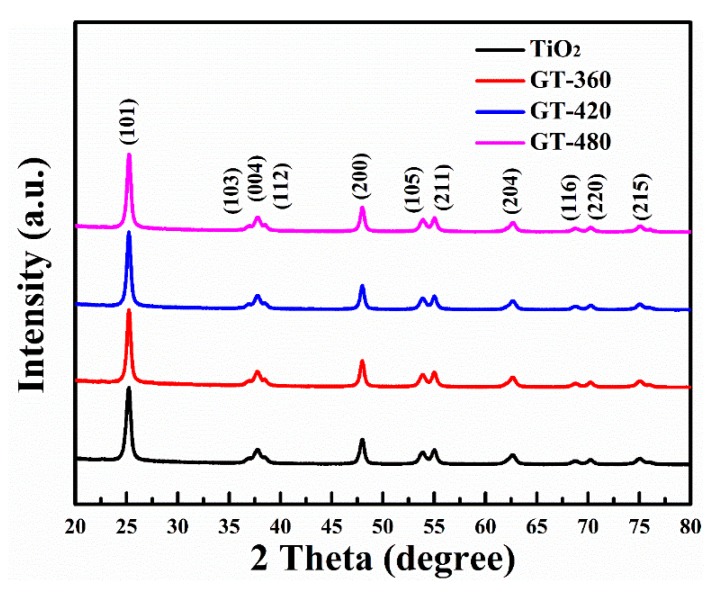
XRD diffraction patterns of gray TiO_2−x_ samples prepared with different glow discharge powers, and black line refers to anatase TiO_2_.

**Figure 4 nanomaterials-10-00342-f004:**
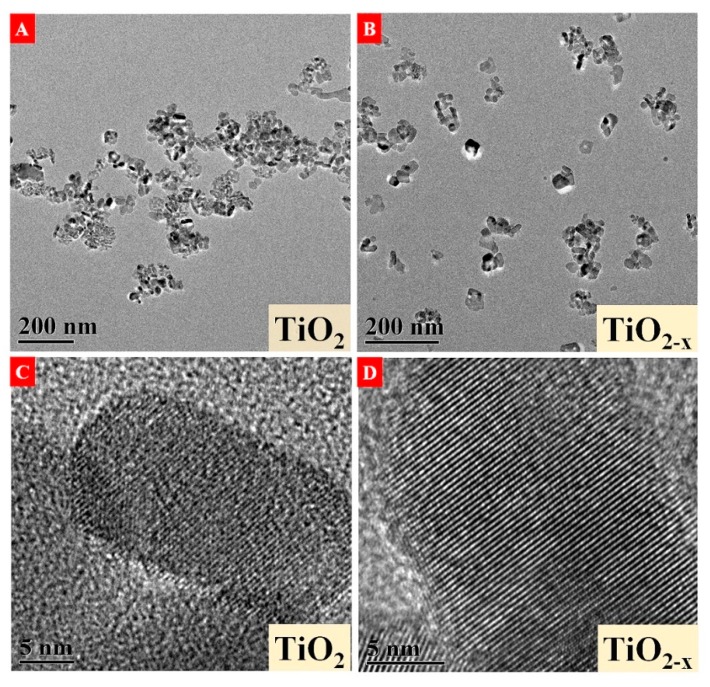
TEM images of (**A**,**C**) pristine TiO_2_ and (**B**,**D**) gray TiO_2−x_.

**Figure 5 nanomaterials-10-00342-f005:**
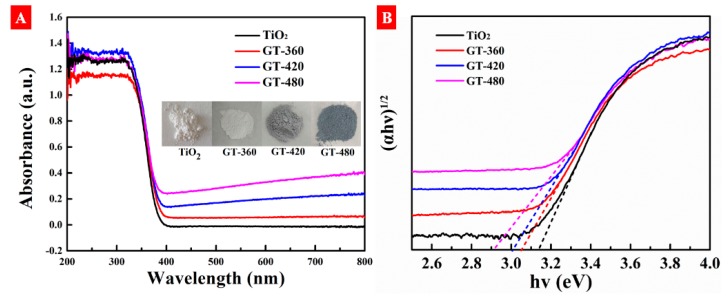
(**A**) UV–Vis diffuse reflectance spectra of as-prepared samples and the digital photos shown in the inset. (**B**) The plot of (αhν)^1/2^ versus hν using the Kubelka–Munk function.

**Figure 6 nanomaterials-10-00342-f006:**
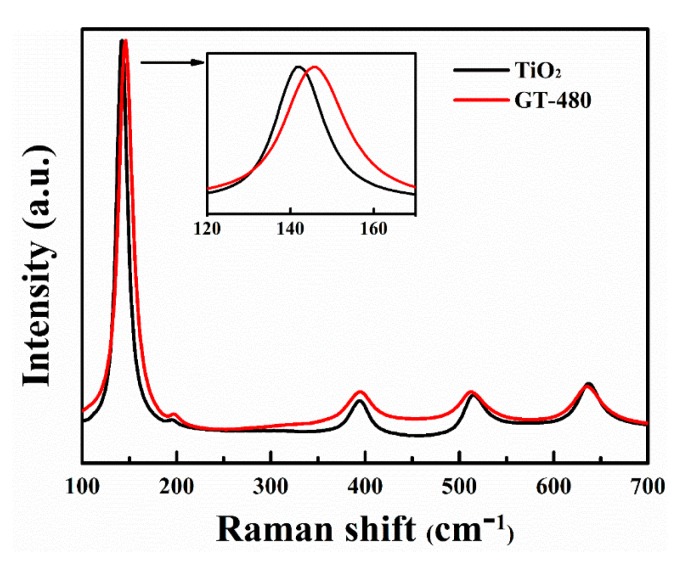
Raman spectra of pristine TiO_2_ and gray TiO_2−x_. The inset shows an enlarged view of E_g_ mode of pristine TiO_2_ and gray TiO_2−x_.

**Figure 7 nanomaterials-10-00342-f007:**
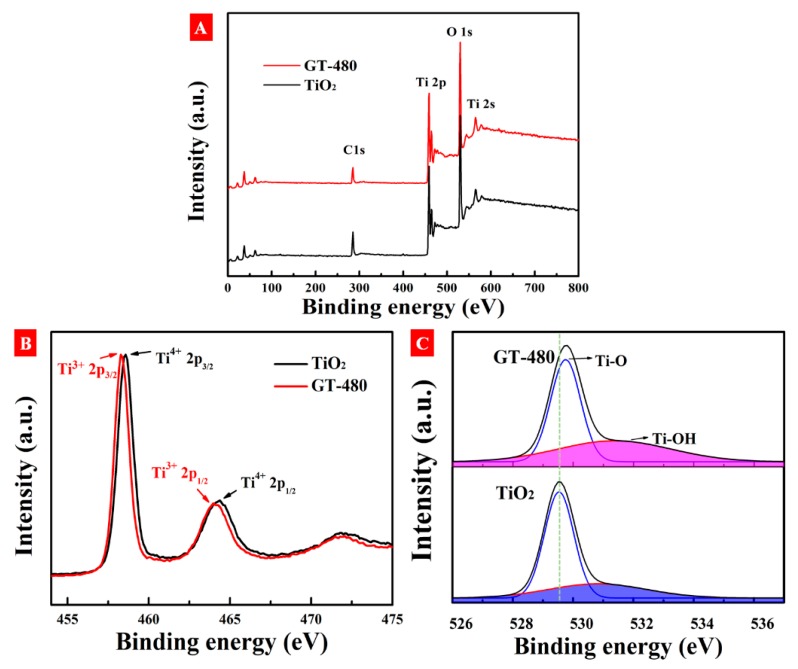
(**A**) XPS survey spectra, (**B**) Ti 2p XPS spectra, and (**C**) O 1s XPS spectra of samples.

**Figure 8 nanomaterials-10-00342-f008:**
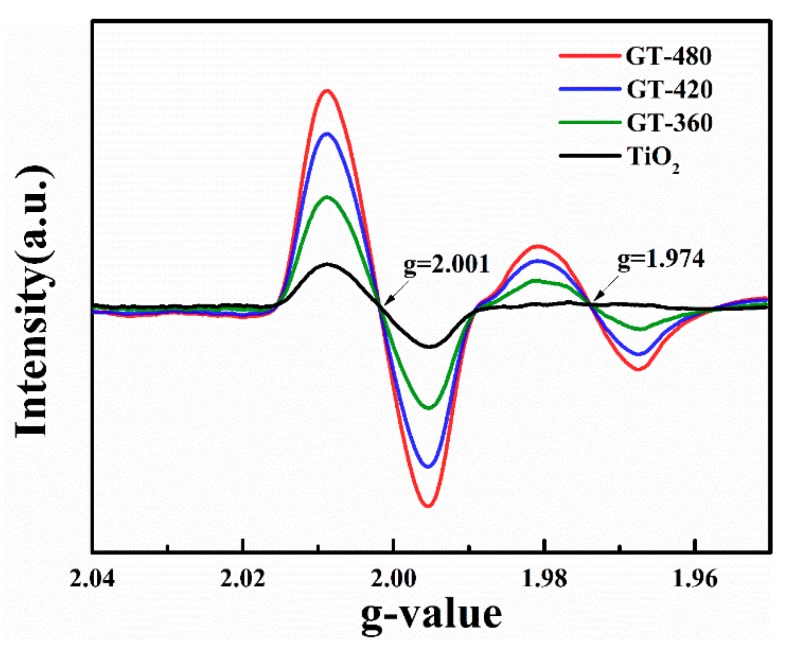
EPR(Electron paramagnetic resonance) spectra of the as-synthesized gray TiO_2−x_ samples obtained with different plasma discharge powers.

**Figure 9 nanomaterials-10-00342-f009:**
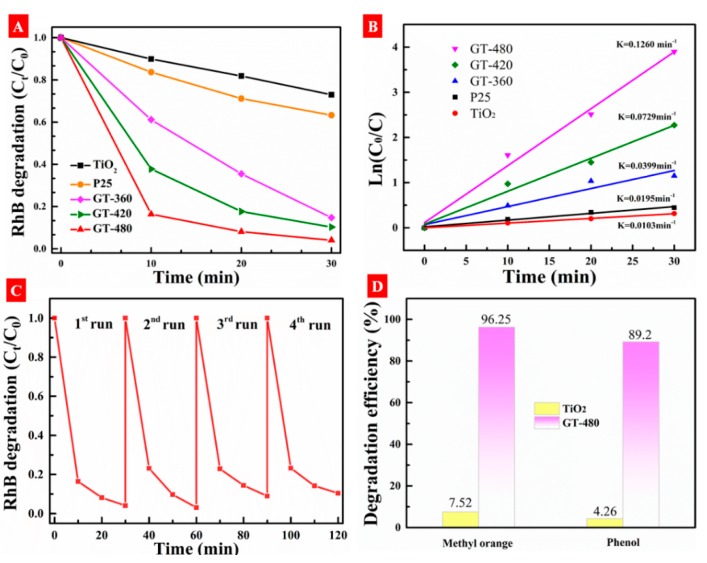
(**A**) Photocatalytic degradation curves of RhB(Rhodamine B) over different photocatalysts under visible-light irradiation. (**B**) The pseudo-first-order reaction rate constant for all samples. (**C**) Recycling test results using GT-480 as the photocatalyst. (**D**) Photodegradation rates of methyl orange and phenol with TiO_2_ and GT-480 under visible-light irradiation.

**Table 1 nanomaterials-10-00342-t001:** Comparison of rhodamine B photodegradation efficiency over gray TiO_2−x_ with reported colored titania and highly efficient photocatalysts.

Photocatalyst	Synthesis Method	Light Source	C_RhB_	W_Cat_	Performance	Ref.
Gray TiO_2_	Liquid-plasma hydrogenation of anatase TiO_2_	Visible light 300 W	50 mL RhB, 20 ppm	50 mg	T_90%_ = 20 min	This work
Black TiO_2_	Hydrogen plasma assisted chemical vapor deposition	Solar light 50 W	50 mL RhB, 2 mg/L	10 mg	T_90%_ = 20 min	[44]
Blue TiO_2_	Hydrothermal treatment and annealing in argon environment	Visible light 350 W	30 mL RhB, 10 mg/L	20 mg	T_90%_ = 150 min	[45]
Blue TiO_2_	Solvothermal reaction with TiCl_3_ solution	Visible light 500 W	50 mL RhB, 10 mg/L	100 mg	T_90%_ = 110 min	[46]
Black TiO_2_	Ultraviolet light irradiation and low temperature annealing	Solar light 500 W	40 mL RhB, 4×10^−5^ M	10 mg	T_90%_ = 130 min	[47]
Gray TiO_2_	Anodization and liquid plasma hydrogenation	Visible light 300 W	50 mL RhB, 20 ppm	50 mg	T_90%_ = 70 min	[25]
Brown TiO_2_	Noble metal deposited on defective TiO_2−x_	Visible light 300 W	10 uM RhB, 80 mL	80 mg	T_90%_ = 120 min	[48]
Gray TiO_2_	Hydrogen plasma-treated nanoporous TiO_2_	Solar light 150 W	3 mg/mL RhB,	0.5 g/L	T_90%_ = 90 min	[49]
Black TiO_2_	Reduced by NaBH_4_ under argon atmosphere.	Solar light 300 W	50 mL RhB, 5 ppm	50 mg	T_90%_ = 50 min	[50]
Ar-TiO_2_	Argon plasma treatments in a dielectric barrier discharge	Visible light 300 W	100 mL RhB 10 mg/L	50 mg	T_90%_ = 90 min	[51]
Graphene-P25	Hydrothermal reaction of graphene and P25	Solar light 100 W	100 mL RhB, 0.03 mmol/L	6.3 mg	T_90%_ = 15 min	[52]
Pt-doped nanoporous TiO_2_	Sol-gel hydrothermal and annealing under H_2_ atmosphere	320–500 nm 220 W	10 mL RhB, 10 ppm	3 mg	T_90%_ > 180 min	[53]

C_RhB_: Initial RhB concentration; W_cat_: Catalyst usage; and T_90%_: Time 90% conversion of RhB; Ref.: Reference.

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
