# Peer review of "Liquid-Plasma Hydrogenated Synthesis of Gray Titania with Engineered Surface Defects and Superior Photocatalytic Activity"

_nanomaterials, 2020, doi:10.3390/nano10020342_

Round 1
Reviewer 1 Report
Author Heping Zeng and his coauthors are describe, “Liquid-plasma hydrogenated synthesis of gray titania 3 with engineered surface defects and superior 4 photocatalytic activity”. These work are really interesting, I recommended accepted after revision.
Abstract should be revised to journal format and added resultant values in abstract. It like a general abstract.
What is main requirements for choosing materials for photocatalyst activities?. Why TiO2 selected?
Author should briefly explained the experimental section and reaction mechanism.
In Characterization, Author provided instrumental details.
In XRD, authors missing peak 112, and 116??
Author mentioned, “full width at half maximum and height (FWMH) as well as the 122 area of the diffraction peak at 101 facet are unchanged after plasma treatment…” how to calculated area of the diffraction peak?
There are any relation in plasma treatment and plasma treatments?.
Conclusion should be revised.?
In Fig. 5A space between wavelength and nm unit. And Figure 7, Binding energy for Binding Energy
Reviewer 2 Report
The manuscript titled "Liquid-plasma hydrogenated synthesis of gray titania with engineered surface defects and superior photocatalytic activity" authors F. Zhang et al. deals with the use of liquid plasma treatment under low temperature and atmospheric pressure to modify commercially available TiO2 (P25) into gray TiO2. The bright liquid plasma was generated on the surface of metallic titanium electrode by applying high voltage pulses with high frequency. The approach is interesting considering particularly the possibility to treat high amount of pristine TiO2 leading to a mass production of gray TiO2. The material was then well characterized by XRD, TEM, UV-Vis, Raman, XPS and EPR wich allow to complete comprehension of the structural modifications occurred at the pristine TiO2 material. Moreover, the authors pointed out a significant improvement of the photocatalytic behavior of the material after the plasma treatment under visible light illumination.
I think the paper is an interesting paper for the scientific community working on photocatalysis. My only doubt concerns the possible contamination of the pristine material by the titanium cathode and/or Pt anode.
The authors discussed the role of the plasma to produce hydrogen environment (deriving from water dissociation) at/near the plasma region. Considering the high temperature involved in the process (2000-3000K) could the authors exclude some possible modification of the material by Ti species deriving from the Titanium cathode used to generate the plasma? The authors must properly discuss this aspect into the manuscript.
Moreover, in order to exclude some possible modification of the material deriving from Pt anode, a quantitative elemental analysis by ICP-OES or ICP-MS of the synthesized material.
Round 2
Reviewer 1 Report
Accept from my side.
Reviewer 2 Report
The manuscript can be now accepted for publication.